# Dynamical network analysis reveals key microRNAs in progressive stages of lung cancer

Chao Kong[1,2,3], Yu-Xiang Yao[3], Zhi-Tong Bing[4,5,6], Bing-Hui Guo[7], Liang Huang[3], Zi-Gang Huang[1,2]*, Ying-Cheng Lai[8,9]

**1** The Key Laboratory of Biomedical Information Engineering of Ministry of Education, Institute of Health and Rehabilitation Science, School of Life Science and Technology, Xi'an Jiaotong University, The Key Laboratory of Neuro-informatics & Rehabilitation Engineering of Ministry of Civil Affairs, Xi'an, Shaanxi, P. R. China, **2** National Engineering Research Center for Healthcare Devices. Guangzhou, Guangdong, P.R. China, **3** Institute of Computational Physics and Complex Systems, School of Physical Science and Technology, Lanzhou University, Lanzhou, China, **4** Evidence Based Medicine Center, School of Basic Medical Science of Lanzhou University, Lanzhou, China, **5** Key Laboratory of Evidence Based Medicine and Knowledge Translation of Gansu Province, Lanzhou, China, **6** Department of Computational Physics, Institute of Modern Physics, Chinese Academy of Sciences, Lanzhou, China, **7** Beijing Advanced Innovation Center for Big Data and Brain Computing, LMIB and School of Mathematics and System Sciences, Beihang University, Beijing, China, **8** School of Electrical, Computer and Energy Engineering, Arizona State University, Tempe, Arizona, United States of America, **9** Department of Physics, Arizona State University, Tempe, Arizona, United States of America

* huangzg@xjtu.edu.cn

**Data Availability Statement:** All relevant data are within the manuscript and its Supporting Information files and in the Zenodo repository (http://doi.org/10.5281/zenodo.3753914).

## Abstract

Non-coding RNAs are fundamental to the competing endogenous RNA (CeRNA) hypothesis in oncology. Previous work focused on static CeRNA networks. We construct and analyze CeRNA networks for four sequential stages of lung adenocarcinoma (LUAD) based on multi-omics data of long non-coding RNAs (lncRNAs), microRNAs and mRNAs. We find that the networks possess a two-level bipartite structure: common competing endogenous network (CCEN) composed of an invariant set of microRNAs over all the stages and stage-dependent, unique competing endogenous networks (UCENs). A systematic enrichment analysis of the pathways of the mRNAs in CCEN reveals that they are strongly associated with cancer development. We also find that the microRNA-linked mRNAs from UCENs have a higher enrichment efficiency. A key finding is six microRNAs from CCEN that impact patient survival at all stages, and four microRNAs that affect the survival from a specific stage. The ten microRNAs can then serve as potential biomarkers and prognostic tools for LUAD.

## Author summary

Lung cancer is the leading cause of cancer-related human deaths worldwide. Lung adenocarcinoma is one of the most common subtypes, and has more pronounced genomic variations than other lung cancer subtypes. A milestone discovery in cancer research is the roles played by non-coding RNAs which have been identified as the oncogenic drivers

**Funding:** ZGH acknowledges supports from NNSF of China under Grants (Nos. 11975178, and 61431012), Natural Science Basic Research Plan in Shaanxi Province of China (Program No. 2020JM-058), Fundamental Research Funds for the Central Universities (sxzd022020012), and support of K. C. Wong Education Foundation. LH acknowledges supports from NNSF of China under Grants Nos. 11775101, and 11422541. YCL would like to acknowledge support from the Vannevar Bush Faculty Fellowship program sponsored by the Basic Research Office of the Assistant Secretary of Defense for Research and Engineering and funded by the Office of Naval Research through Grant No. N00014-16-1-2828. BHG acknowledges support from Artificial Intelligence Project (2018AAA0102301). The funders had no role in study design, data collection and analysis, decision to publish, or preparation of the manuscript.

**Competing interests:** The authors have declared that no competing interests exist.

and tumor suppressors. In cancer development, non-coding RNAs form an inseparable unity of RNA-level regulating networks in the intracellular environment, and the dynamical interplay and competition among different types of RNAs are playing a pivotal role. We have developed a quantitative approach to reconstructing the the mutual regulation networks of RNAs for the progressive stages of lung adenocarcinoma at the post-transcriptional level. Our analysis revealed the emergence of two characteristically distinct types of networks that possess a two-level bipartite structure, and we uncovered a number of key genes that affect or even determine the survival of patients at each stage. Our work establishes a more comprehensive gene-data analysis framework than previous ones, not only providing a tool to probe more deeply into the mechanisms of cancer evolution than previously possible but also having the potential to lead to more effective biomarkers and drug targets for lung cancer.

## Introduction

Lung cancer is the leading cause of cancer-related human deaths worldwide [1]. Approximately 85% of the lung cancer cases can be classified as non small-cell lung cancer (NSCLC), among which lung adenocarcinoma (LUAD) is one of the most common subtypes [2]. The mechanisms behind cancer evolution are extremely complex, impeding accurate and reliable prognosis as well as effective treatment [3]. A standard existing approach to monitoring tumor progress and detecting/ascertaining the underlying mechanism is mRNA transcriptomics, which has led to a large number of significant prognostic biomarkers and therapeutic targets [4–6]. When combined with other omics information, such as microRNA, methylation and epigenetic data, mRNA transcriptomics has provided valuable insights into the mechanisms of lung cancers [7–10].

A milestone discovery in cancer research is the roles played by non-coding RNAs (ncRNAs). In the general developmental and disease contexts, at the mRNA level, ncRNAs have been found in key regulators of physiological functions [11–15]. Particularly relevant to cancer, ncRNAs have been identified as the oncogenic drivers and tumor suppressors in major cancer types [16]. The discovery of the pivotal role of ncRNAs in cancer has led to the paradigmatic, competing endogenous RNA (CeRNA) hypothesis [17–19]: in cancer emergence and development, microRNAs, mRNAs, and ncRNAs form an inseparable unity of RNA-level regulating network in the intracellular environment, collectively known as the CeRNA network. For example, it has been known [8, 18] that RNA-induced silencing complex (RISC) can be produced through binding of microRNA response elements (MREs) at RNAs 3'UTR, causing inactivation of the target mRNA, but this mechanism is also present in non-coding RNAs, where the target RNA usually has multiple MREs. There has been increasing evidence for the fundamental roles played by CeRNAs in biological systems [16, 18] and, as a result, studying cancer-related CeRNA networks based on RNA-sequence expression data has gained momentum. For example, dysregulated CeRNA-CeRNA interactions in the CeRNA networks of LUAD were analyzed [20, 21], suggesting that the gain or loss of CeRNAs can be used in functional analysis and as potential diagnostic biomarkers. The difference in the expression profiles between early and late stages in CeRNA networks of LUAD was noticed and some LUAD specific, long non-coding RNAs (lncRNAs) with their functional enrichment and clinical features were uncovered [22]. The possible roles played by lncRNAs through CeRNA networks in other types of cancer such as liver cancer and papillary thyroid cancer were also studied [5].

In existing studies of CeRNA networks, a commonly practiced methodology is to identify some differentially expressed lncRNAs, mRNAs and microRNAs based on absolute fold changes and values of the false positive ratio. Consequently, information used to construct the underlying CeRNA networks and to reveal the network functions with clinical implications is directly from data bases without any dynamical ingredient. The resulting CeRNA networks are thus simply a combination of different kinds of RNAs, whereas the dynamical interplay and competition among different types of RNAs were completely ignored. To remedy this deficiency has motivated our work.

We hold the belief that cancer development and evolution are fundamentally a dynamic process. Manifested in the underlying CeRNA networks, it is unlikely that the intrinsic interactions, interplay and the network structure remain static during cancer development. To better understand cancer and to identify more effective biomarkers, the dynamical aspects of the CeRNA networks must be taken into account. This is in line with the field of network physiology [23–27], where transitions and reorganization occur in the networks of physiological/organ systems in the human body on larger spatial and temporal scales, which can be constructed using readily accessible continuous time series. In the CeRNA network analysis, it is infeasible to collect RNA data in continuous time. Nonetheless, with the presently available gene data, we are able to incorporate the time axis into the analysis but only for a limited set of time intervals. In particular, we focus on the four stages of LUAD progression to construct, based on both RNA expression and clinical data of LUAD from the cancer genome atlas (TCGA), the lncRNA-microRNA-mRNA CeRNA network for each stage. For any given stage, our quantitative approach to reconstructing the CeRNA network consists of analyzing differentially expressed RNAs, matching microRNA targets by base complementary pairing, and selecting the negative correlation by invoking the CeRNA hypothesis. We then calculate the fold changes and the average expression level of RNAs in the CeRNA networks for the four stages and carry out Gene Ontology (GO) and Kyoto Encyclopedia of Genes and Genomes (KEGG) pathway enrichment analyses to validate the results. Our analysis reveals the emergence of two characteristically distinct types of networks that play an important role in the progression of LUAD from stage I to stage IV: one is common to all four stages, which we name as the common competing endogenous RNA network (CCEN), and another is unique for each stage, which we call as the unique competing endogenous RNA networks (UCENs). Analyzing the properties of CCEN and UCENs, we uncover a number of key genes that affect or even determine the survival of patients at each stage of LUAD: six microRNAs from the CCEN which affect the survival of LUAD patients at all stages, and four other microRNAs that influence the survival at each specific stage. Our work establishes a more comprehensive gene-data analysis framework than previous ones, not only providing a tool to probe more deeply into the mechanisms of cancer evolution than previously possible but also having the potential to lead to more effective biomarkers and drug targets for LUAD as well as other types of cancer.

## Results

### Reconstructed lncRNA-microRNA-mRNA CeRNA networks and doubly bipartite structure

As described in **Materials and Methods**, to reconstruct the lncRNA-microRNA-mRNA CeRNA networks, we first obtain the differentially expressed RNAs by comparing the gene expression values in the LUAD samples with those from the normal samples of lncRNAs, microRNAs and mRNAs. We then apply the principle of base complementary pairing to match the microRNAs with lncRNAs and mRNAs, all differentially expressed, leading to values of the correlation between microRNA-lncRNA and microRNA-mRNA nodes. Retaining

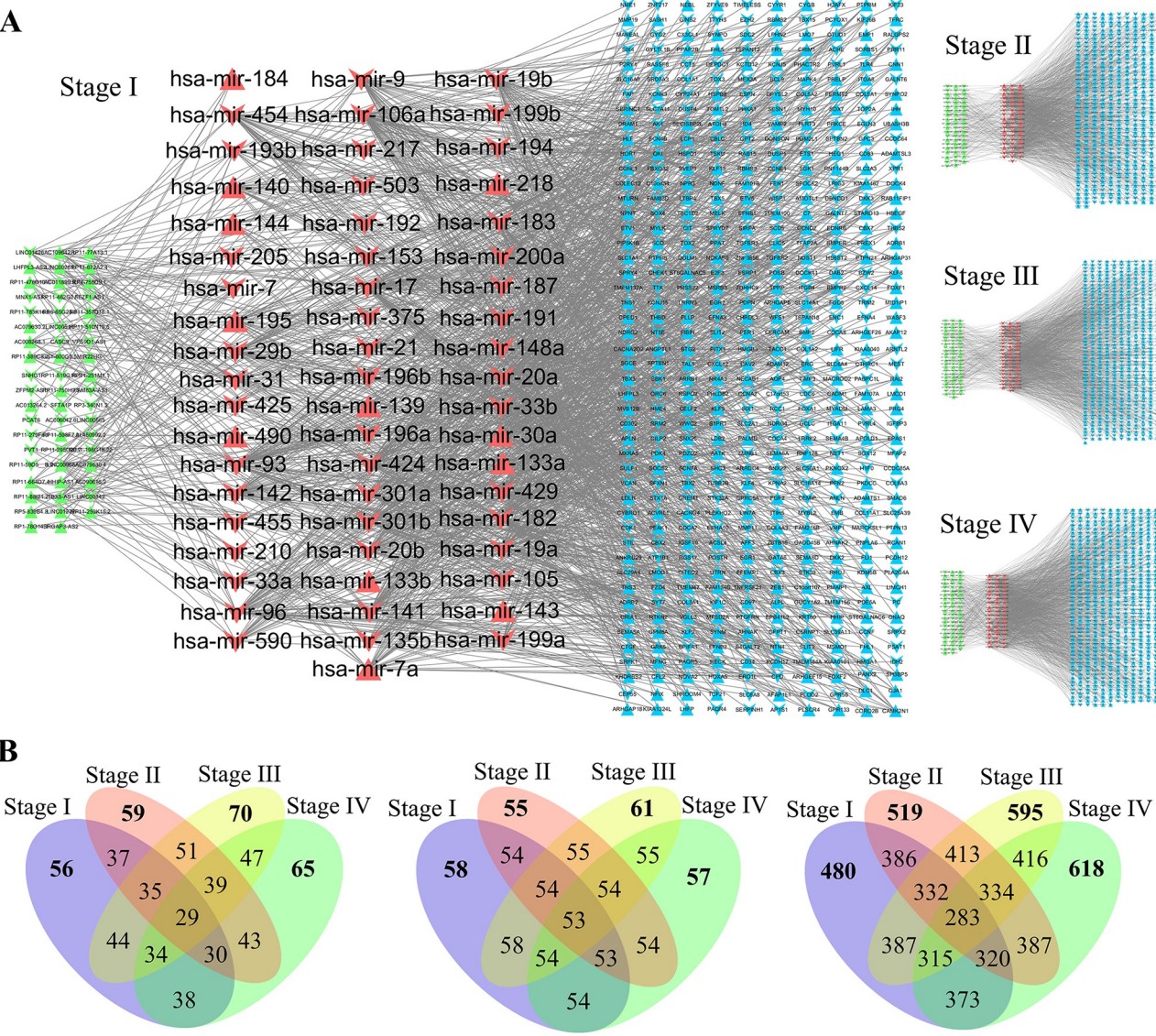

**Fig 1. Doubly bipartite or mutualistic structure of the lncRNA-microRNA-mRNA CeRNA networks.** (A) Reconstructed lncRNA-microRNA-mRNA CeRNA networks for the four stages of LUAD, where the green, red, and blue nodes indicate lncRNAs, microRNAs, and mRNAs, respectively. The edges represent the mutual regulation between microRNAs and lncRNAs or between microRNAs and mRNAs, which are from the data bases of microRNA targets. At a larger scale where the clusters of lncRNA, microRNA, and mRNA nodes are viewed as three "supernodes," there is mutualism because the interaction between the lncRNA and mRNA supernodes is through the microRNA supernode. Mutualism also arises at the smaller scale of individual RNA nodes. Especially, the lncRNAs and microRNAs (or the mRNAs and microRNAs) constitute a bipartite network where the interactions between any two nodes of the same color must be through a node of a different color. (B) Venn diagrams illustrating the numbers of nodes in the lncRNA (left), microRNA (middle) and mRNA (right) CeRNA networks of the four stages of LUAD, reflecting the dynamical evolution of the networks.

only the negatively correlated pairs of microRNA-lncRNAs and microRNA-mRNAs, we obtain the final CeRNA networks. (S1 Table presents more details of the network reconstruction process).

Fig 1 presents examples of the reconstructed lncRNA-microRNA-mRNA CeRNA networks for the four stages of LUAD. A common feature for the networks in all four states is that direct connections exist only between microRNAs (red nodes) and lncRNAs (green nodes) or between the microRNAs and mRNAs (blue nodes). The connections between lncRNAs and

mRNAs are thus indirect: they occur through the microRNAs. Regarding the lncRNAs, mRNAs, and microRNAs as three entities (or "supernodes"), we see that the connection between the first two supernodes is through the third one, which is characteristic of mutualistic or bipartite type of interactions. The mutualism occurs at a finer scale because the subnetwork of microRNAs-lncRNAs (or that of microRNAs-mRNAs) also has a bipartite structure: there are no direct links among the microRNAs or among the lncRNAs, i.e., any connection between two nodes of the same color must be indirect and through a node of a different color. Such bipartite network structure arises commonly in ecology, e.g., the pollinator-plant mutualistic networks [28–36]. The striking feature here is that the lncRNA-microRNA-mRNA CeRNA networks for all four stages of LUAD possess a doubly bipartite structure: mutualism occurs at two distinct scales.

## Changes in fold-change value and expression

Compared with RNAs from the direct differential expression analysis, the differentially expressed RNAs of the final CeRNA networks obtained through matching the differentially expressed microRNA targets and retaining only those RNA pairs with negative correlation have higher values of the expression of microRNAs and fold change, as shown in Fig 2. Our

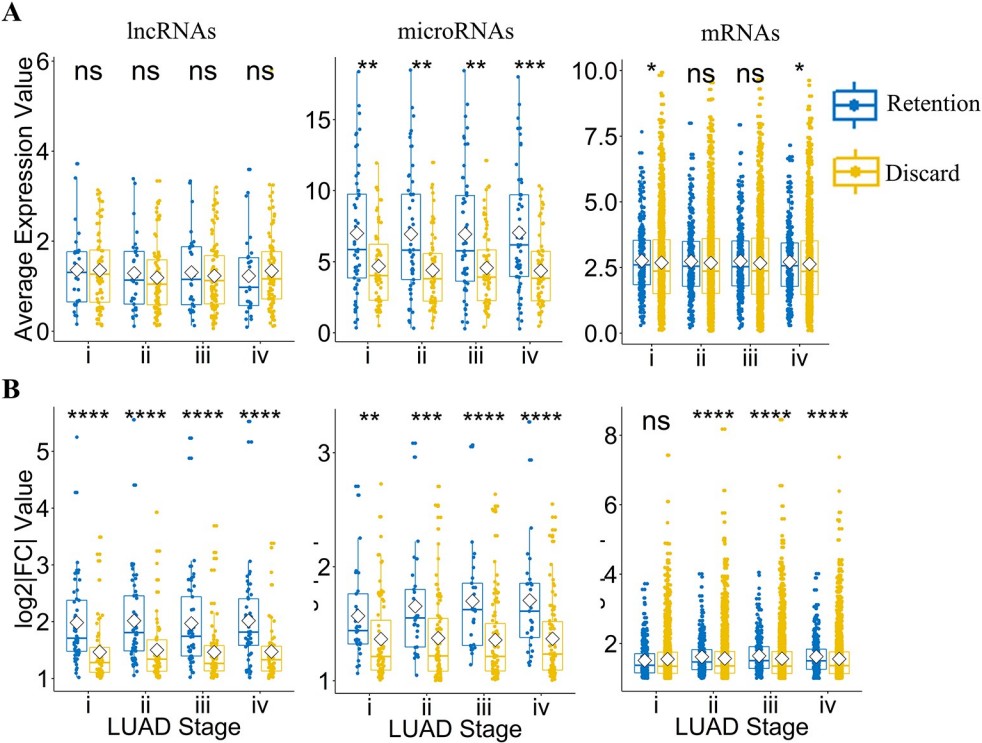

**Fig 2. Gene expression and fold-change values of lncRNA, microRNA and mRNA nodes in the reconstructed CeRNA networks of the four stages of LUAD.** The two-sample Wilcoxon test (also known as "Mann-Whitney" test) [37, 38] is applied to vectors of the RNA expression data, with the following notations for statistical significance: ns—$p > 0.05$; *—$p \leq 0.05$; **—$p \leq 0.01$; ***—$p \leq 0.001$; ****—$p \leq 0.0001$. The blue group "Retention" represents RNAs selected through analyzing differentially expressed (DE) RNAs, matching microRNA targets, and selecting the negative correlation in the CeRNA networks. For comparison, the yellow group "Discard" describes RNAs processed only through the differential expression analysis, which are not used in the construction of the CeRNA networks. The hypothesis tests between the "Discard" and "Retention" groups are of the Wilcoxon type. (A) Gene expression values of lncRNAs, microRNAs and mRNAs in the reconstructed CeRNA networks of the four stages of LUAD. (B) The corresponding fold-change values of lncRNAs, microRNAs and mRNAs.

**A**

| | Stage all | Stage I | Stage II | Stage III | Stage IV | |
|---|---|---|---|---|---|---|
| | 29.79 | 21.26 | 20.14 | 24.18 | 23.15 | blood vessel development |
| | 26.75 | 22.97 | 21.97 | 21.03 | 20.85 | tissue morphogenesis |
| | 23.58 | 14.99 | 16.14 | 18.53 | 17.08 | regulation of cell adhesion |
| | 19.62 | 15.89 | 14.8 | 16.63 | 10.48 | extracellular matrix organization |
| | 19.49 | 12.4 | 12.21 | 11.27 | 12.85 | cell-substrate adhesion |
| | 17.25 | 8.11 | 9.09 | 13.29 | 10.03 | negative regulation of cell differentiation |
| | 16.71 | 12.42 | 11.74 | 13.29 | 13.15 | response to growth factor |
| | 16.41 | 13.55 | 11.89 | 14.18 | 10.06 | mesenchyme development |
| | 16.28 | 10.13 | 11.86 | 14.28 | 10.32 | developmental growth |
| | 13.21 | 12.33 | 10.03 | 12.54 | 12.39 | negative regulation of cell proliferation |
| | 10.6 | 6.74 | 6.18 | 8.86 | 7.88 | negative regulation of cell adhesion |
| | 9.65 | 6.47 | 5.85 | 9.65 | 5.96 | regulation of cell-cell adhesion |
| | 9.56 | 6.47 | 5.85 | 8.47 | 6.46 | positive regulation of cell cycle |
| | 9.32 | 6.12 | 7.62 | 9.32 | 7.63 | developmental growth involved in morphogenesis |
| | 8.29 | 5.03 | 5.98 | 4.66 | 7.96 | positive regulation of cell death |
| | 7.61 | 3.1 | 4.47 | 3.11 | 7.61 | negative regulation of cell cycle |
| | 6.28 | 4.19 | 3.93 | 5.77 | 4.82 | substrate adhesion-dependent cell spreading |
| | 6.25 | 3.84 | 4.39 | 3.21 | 3.82 | vasculogenesis |
| | 6.06 | 5.29 | 6.06 | 3.73 | 3.63 | coronary vasculature development |
| | 5.78 | 2.18 | 2.65 | 5.17 | 2.24 | positive regulation of cell division |
| | 5.49 | 4.25 | 4.04 | 5.49 | 2.05 | lung morphogenesis |
| | 5.18 | 5.02 | 2.66 | 4.88 | 4.02 | negative regulation of developmental growth |

**B**

| | Stage all | Stage I | Stage II | Stage III | Stage IV | |
|---|---|---|---|---|---|---|
| | 12.17 | 2.98 | 4.75 | 7.77 | 10.27 | Pathways in cancer |
| | 8.54 | 3.72 | 4.11 | 4.23 | 8.54 | Cell cycle |
| | 7.5 | 2.83 | 2.89 | 5.81 | 3.24 | PI3K-Akt signaling pathway |
| | 6.69 | 4.44 | 4.18 | 6.11 | <2.00 | ECM-receptor interaction |
| | 6.63 | 3.73 | 3.4 | 6.26 | 2.71 | Focal adhesion |
| | 5.68 | <2.00 | 3.33 | 4.14 | 4.96 | HTLV-I infection |
| | 5.6 | 2.08 | 2.54 | 2.83 | 5.6 | Gap junction |
| | 5.38 | 3.56 | 3.83 | 3.26 | 5.38 | Transcriptional misregulation in cancer |
| | 5.05 | 5.05 | 3.94 | 4.32 | 4.19 | p53 signaling pathway |
| | 5.04 | <2.00 | 2.02 | 3.65 | 2.86 | Small cell lung cancer |
| | 4.91 | 4.91 | 3.86 | 4.88 | <2.00 | Protein digestion and absorption |
| | 4.83 | 2.11 | 2.81 | 3.78 | 4.13 | Proteoglycans in cancer |
| | 4.61 | <2.00 | <2.00 | 4.15 | 2.51 | Rap1 signaling pathway |
| | 4.48 | 2.14 | 2.65 | 2.88 | 4.48 | MicroRNAs in cancer |
| | 3.93 | <2.00 | <2.00 | 3.14 | 2.39 | EGFR tyrosine kinase inhibitor resistance |
| | 3.18 | <2.00 | <2.00 | <2.00 | 3.1 | Circadian entrainment |

**Fig 3. Gene enrichment analysis of mRNAs in the CeRNA networks of four stages of LUAD.** The first column in the heat map indicates the results of enrichment analysis. The second to fifth columns represent the results of analysis from the first to the fourth LUAD stage, respectively. The values of the heat map are those of $-\log_{10} P$ from the enrichment analysis. The hypergeometric distribution test is used to calculate the P-value. (A) Results of gene ontology enrichment analysis of various biological processes. (B) Results of KEGG enrichment analysis.

construction of the CeRNA networks thus result in more deviated RNAs relative to normal sample expression of LUAD.

## Validation of network reconstruction: Enrichment analysis of mRNAs in CeRNA networks

Are the lncRNA-microRNA-mRNA CeRNA networks so constructed meaningful? To address this question, we perform gene ontology and KEGG pathway gene enrichment analysis of mRNAs in the CeRNA networks at the four stages of LUAD, with results in Fig 3. We find that the mRNAs in the CeRNA networks at all four stages are strongly correlated with blood vessel development ($-\log_{10} P > 20$), tissue morphogenesis ($-\log_{10} P > 20$) and regulation of cell adhesion ($-\log_{10} P > 15$), where $P$ is the P-value (Materials and methods). Strong correlation

($-\log_{10} P > 10$) also exists between the mRNAs and factors such as extracellular matrix organization, cell-substrate adhesion, response to growth factor, mesenchyme development, developmental growth and negative regulation of cell proliferation. Remarkably, beyond the "static" information provided by the conventional gene ontology analysis of the four stages of LUAD, our CeRNA networks give rise to a dynamic scenario for tumor progression as the TNM stage deteriorates (see Materials and methods for the meaning of TNM). For example, mRNAs are more correlated with coronary vasculature development and less correlated with substance-dependent cell spreading in the early than the late two stages. In addition, mRNAs are more correlated with positive regulation of cell death and negative regulation of cell cycle ($-\log_{10} P > 7$) and less correlated with lung morphogenesis ($-\log_{10} P < 6$) in the late stages than the early three stages of LUAD.

The KEGG pathway enrichment analysis reveals that the mRNAs associated with pathway in cancer ($-\log_{10} P = 2.98, 4.75, 7.77$, and $10.2$ for the four stages, respectively) and cell cycle ($-\log_{10} P = 3.72, 4.11, 4.23$, and $8.54$ for the four stages, respectively) have stronger correlation in early than late TNM stages of LUAD. Additionally, some cancer-related pathways in the late stages are more significant than in the early stages, such as PI3K-Akt signaling pathway, focal adhesion, gap junction, transcriptional misregulation in cancer, proteoglycans in cancer, Rap1 signaling pathway, microRNAs in cancer, EGFR tyrosine kinase inhibitor resistance, and circadian entrainment. On the contrary, ECM-receptor interaction as well as protein digestion and absorption are less relevant to the LUAD late stages than to the early stages.

Taken together, the enrichment analysis of mRNAs reveals that our reconstructed lncRNA-microRNA-mRNA CeRNA networks exhibit a close correspondence to the development of LUAD and deterioration of physiological indicators from stage I to stage IV, validating our reconstruction method.

## CCEN and UCEN analysis

To better understand the relationship between CeRNA networks and TNM stages of LUAD, we extract the CCEN and UCENs from the lncRNA-microRNA-mRNA CeRNA networks, as shown in Fig 4.

Fig 5A shows the results of gene ontology enrichment analysis. We see that the mRNAs of the UCENs are correlated with blood vessel development, tissue morphogenesis, regulation of cell adhesion, extracellular matrix organization, cell-substrate adhesion and regulation of growth ($-\log_{10} P > 5$). Consistent with the results of enrichment analysis of the mRNAs in the CeRNA networks (Fig 3), the UCENs can also distinguish the biological characteristics between early and later stages of the cancer. In particular, the UCENs from the late stages are more correlated with regulation of growth, negative regulation of cell adhesion, positive regulation of cell death, substrate adhesion-dependent cell spreading, regulation of cell division and vascular process in circulatory system than those in the early stage. However, extracellular matrix organization and mesenchyme development in the fourth stages do not exhibit significant changes.

From the KEGG pathway enrichment analysis (Fig 5B), we find that the CCEN network is relevant to pathways in cancer, small cell lung cancer, PI3K-Akt signaling pathway, ECM-receptor interaction, focal adhesion and cell cycle, while all these pathways belonging to the later two stages show a stronger correlation than those in the early two stages.

## Enrichment efficiency of microRNA-linked mRNAs in UCENs

Our reconstructed CeRNA networks, UCEN, and CCEN have 868, 585, and 283 mRNAs altogether, respectively. In comparison, in the UCEN there are 201 microRNA-linked mRNAs.

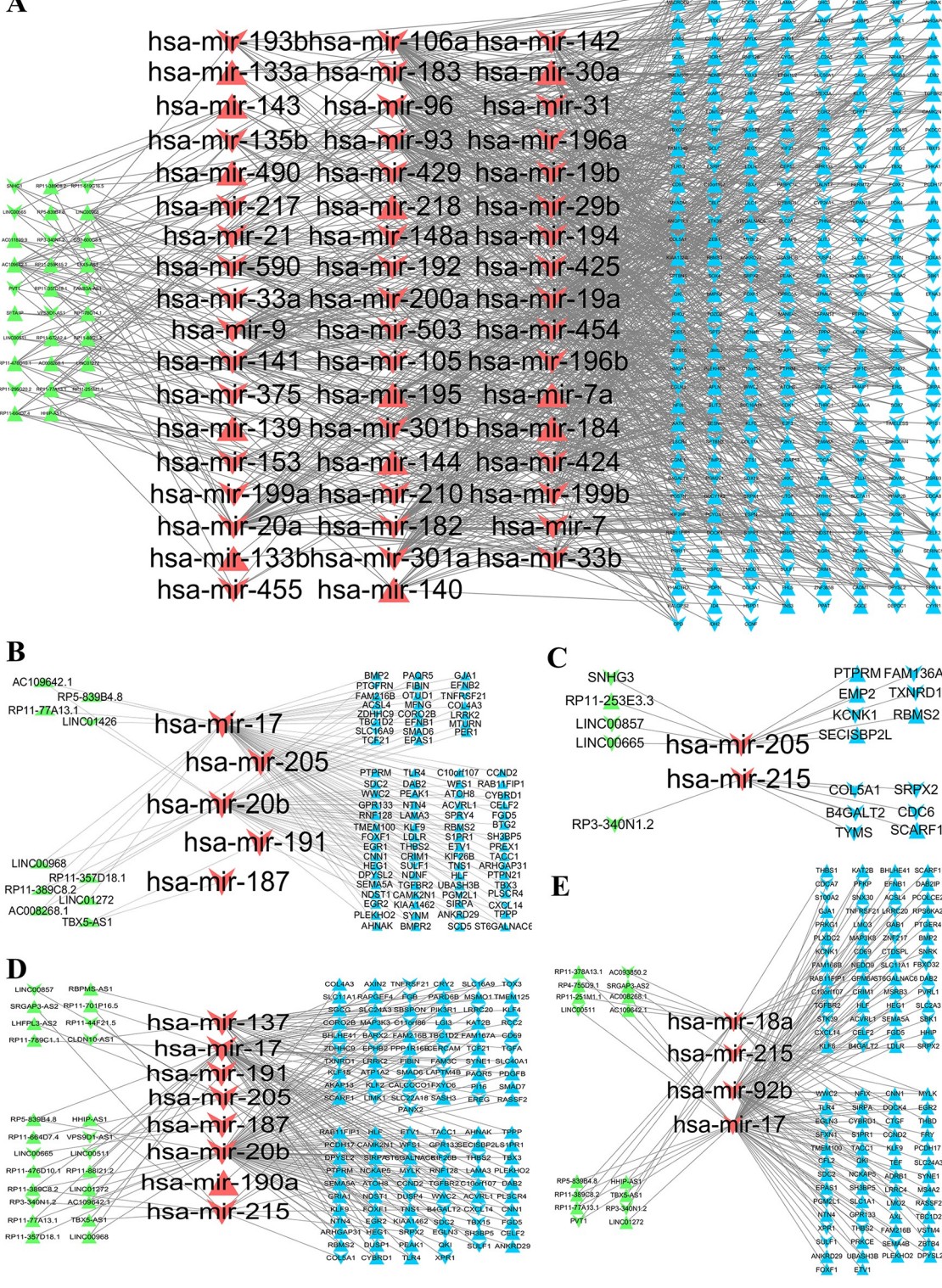

**Fig 4. Representative CCEN and UCENs.** (A) CCEN extracted from the lncRNA-microRNA-mRNA CeRNA networks of the four stages of LUAD. (B-E) UCENs corresponding to the four stages. The green, red, and blue nodes represent lncRNAs, microRNAs, and mRNAs, respectively. The edges represent mutual regulation between microRNA-lncRNA and between microRNA-mRNA from the microRNA target data bases.

**A**

| | All | Common | Stage I | Stage II | Stage III | Stage IV |
|---|---|---|---|---|---|---|
| blood vessel development | 29.79 | 14.76 | 7.16 | 6.31 | 10.07 | 9.23 |
| tissue morphogenesis | 26.75 | 13.87 | 9.56 | 8.65 | 7.99 | 7.9 |
| regulation of cell adhesion | 23.58 | 10.05 | 5.56 | 6.69 | 9 | 7.68 |
| extracellular matrix organization | 19.62 | 5.36 | 11.88 | 10.37 | 12.06 | 5.65 |
| cell-substrate adhesion | 19.49 | 5.52 | 7.65 | 7.32 | 6.27 | 7.84 |
| positive regulation of locomotion | 18.65 | 11.11 | 2.13 | 2.37 | 4.72 | 5.78 |
| negative regulation of cell differentiation | 17.25 | 4.28 | 4.51 | 5.51 | 9.95 | 6.39 |
| response to growth factor | 16.71 | 9.19 | 3.98 | 3.51 | 5 | 4.93 |
| mesenchyme development | 16.41 | 6.1 | 8.16 | 6.31 | 8.56 | 4.56 |
| regulation of growth | 15.66 | 4.1 | 5.93 | 7 | 6.9 | 7.88 |
| negative regulation of cell proliferation | 13.21 | 8.16 | 4.81 | 2.87 | 5.16 | 5.07 |
| pattern specification process | 10.71 | 3.69 | 3.03 | 3.98 | 4.79 | 3.87 |
| gland development | 10.67 | 6.61 | <2.00 | 4.56 | 2.77 | 2.49 |
| negative regulation of cell adhesion | 10.6 | 3.96 | 3.33 | 2.78 | 5.46 | 4.45 |
| rhythmic process | 10.55 | 4.58 | 4.01 | 2.77 | 4.15 | 6.42 |
| positive regulation of cell cycle | 9.56 | 4.78 | 2.31 | <2.00 | 4.27 | 2.47 |
| positive regulation of cell death | 8.29 | 3.7 | <2.00 | 2.89 | <2.00 | 4.87 |
| regulation of cell cycle process | 7.88 | 3.84 | <2.00 | <2.00 | 2.88 | 3.25 |
| regulation of transmembrane transport | 7.72 | <2.00 | 3.32 | 2.17 | 5.13 | 2.45 |
| substrate adhesion-dependent cell spreading | 6.28 | 3.22 | <2.00 | <2.00 | 3 | 2.12 |
| vasculogenesis | 6.25 | <2.00 | 2.44 | 3.02 | <2.00 | 2.36 |
| regulation of cell division | 5.85 | <2.00 | <2.00 | <2.00 | 5.85 | 2.12 |
| vascular process in circulatory system | 5.84 | 3.19 | <2.00 | <2.00 | <2.00 | 2.72 |

**B**

| | All | Common | Stage I | Stage II | Stage III | Stage IV |
|---|---|---|---|---|---|---|
| Pathways in cancer | 12.17 | 3.07 | <2.00 | 2.24 | 5.35 | 8.05 |
| PI3K-Akt signaling pathway | 7.5 | <2.00 | 2.68 | 2.68 | 6.23 | 2.97 |
| Cell cycle | 7.01 | 3.96 | <2.00 | <2.00 | <2.00 | 5.03 |
| ECM-receptor interaction | 6.97 | <2.00 | 5.31 | 4.8 | 6.97 | <2.00 |
| Focal adhesion | 6.63 | <2.00 | 2.89 | 2.45 | 5.67 | <2.00 |
| p53 signaling pathway | 5.84 | 5.84 | <2.00 | <2.00 | <2.00 | <2.00 |
| HTLV-I infection | 5.68 | 2.47 | <2.00 | <2.00 | 2.2 | 3.02 |
| Gap junction | 5.53 | <2.00 | <2.00 | <2.00 | 2.24 | 5.43 |
| Small cell lung cancer | 5.04 | <2.00 | <2.00 | <2.00 | 3.11 | 2.2 |
| Rap1 signaling pathway | 4.7 | <2.00 | <2.00 | <2.00 | 4.7 | 2.56 |
| Transcriptional misregulation in cancer | 4.55 | 3.54 | <2.00 | <2.00 | <2.00 | 2.42 |
| EGFR tyrosine kinase inhibitor resistance | 4.13 | <2.00 | <2.00 | <2.00 | 4.13 | 3.09 |
| Protein digestion and absorption | 4.04 | 2.38 | 3.06 | <2.00 | 2.95 | <2.00 |
| Circadian entrainment | 3.18 | <2.00 | <2.00 | <2.00 | <2.00 | 2.65 |

**Fig 5. Gene enrichment analysis of mRNAs in the CCEN and UCENs associated with the four stages of LUAD.** The first column "All" in the heat map presents the results of enrichment analysis in the four LUAD stages, and the column "Common" lists the results of enrichment analysis in the CCEN. The columns "Stage I" to "Stage IV" show the results of mRNAs in the four stages. The value of the heat map is $-\log_{10} P$ from the enrichment analysis. The P-value is calculated using the hypergeometric distribution test. (A) Results from gene ontology enrichment analysis of various biological processes. (B) Results from KEGG enrichment analysis.

We find that they offer the best average enrichment efficiency in blood vessel development, regulation of cell adhesion, circulatory system process, tissue morphogenesis and pathways in cancer, as shown in Fig 6. The microRNA-associated CeRNA networks with higher enrichment efficiency thus exhibit stronger correlation with LUAD development than other types of CeRNA networks.

## MicroRNAs in CCEN with a significantly influence on patient survival

We perform the Kaplan-Meier analysis [39] and find that there are six microRNAs that have a significant effect on patient survival for all stages of LUAD, as shown in Fig 7. They are hsa-

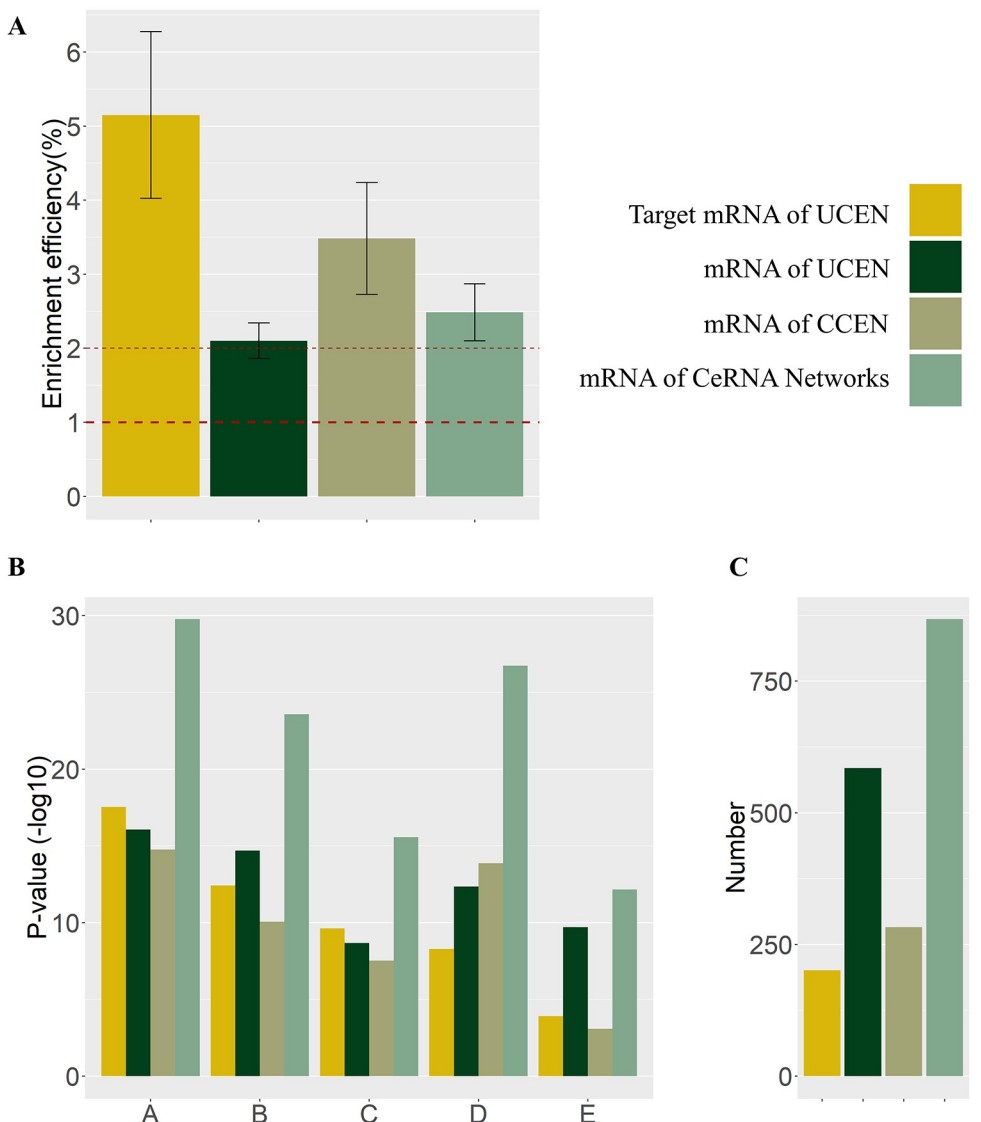

**Fig 6. Results of enrichment efficiency analysis.** (A) Values of enrichment efficiency of the mRNAs associated with microRNAs in the UCEN and those of direct analysis of mRNAs in CCEN, UCEN, and CeRNA networks. (B) P-values of enrichment analysis based on gene ontology and KEGG, where columns A, B, C, D and E correspond to blood vessel development, regulation of cell adhesion, circulatory system process, tissue morphogenesis, and pathways in cancer, respectively. (C) The numbers of mRNAs associated with microRNAs in UCEN and of all mRNAs in the CCEN, UCEN and CeRNA networks.

mir-9, hsa-mir-21, hsa-mir-31, hsa-mir-148a, hsa-mir-195, and hsa-mir-375. For each stage, there is a specific microRNA that affects the survival rate: hsa-mir-19a (P-value, 0.048), hsa-mir-196b (P-value, 0.058), hsa-mir-194 (P-value, 0.061), and hsa-mir-144 (P-value, 0.012) for stages I-IV, respectively, as shown in Fig 8. To further support for this finding, we also calculate the four microRNAs survival curves of LUAD patients at each stage, and find that hsa-mir-19a, hsa-mir-196b, hsa-mir-194 and hsa-mir-144 have little effect on the survival at other stages except for stages I (S1 Fig), II (S2 Fig), III (S3 Fig) and IV (S4 Fig), respectively.

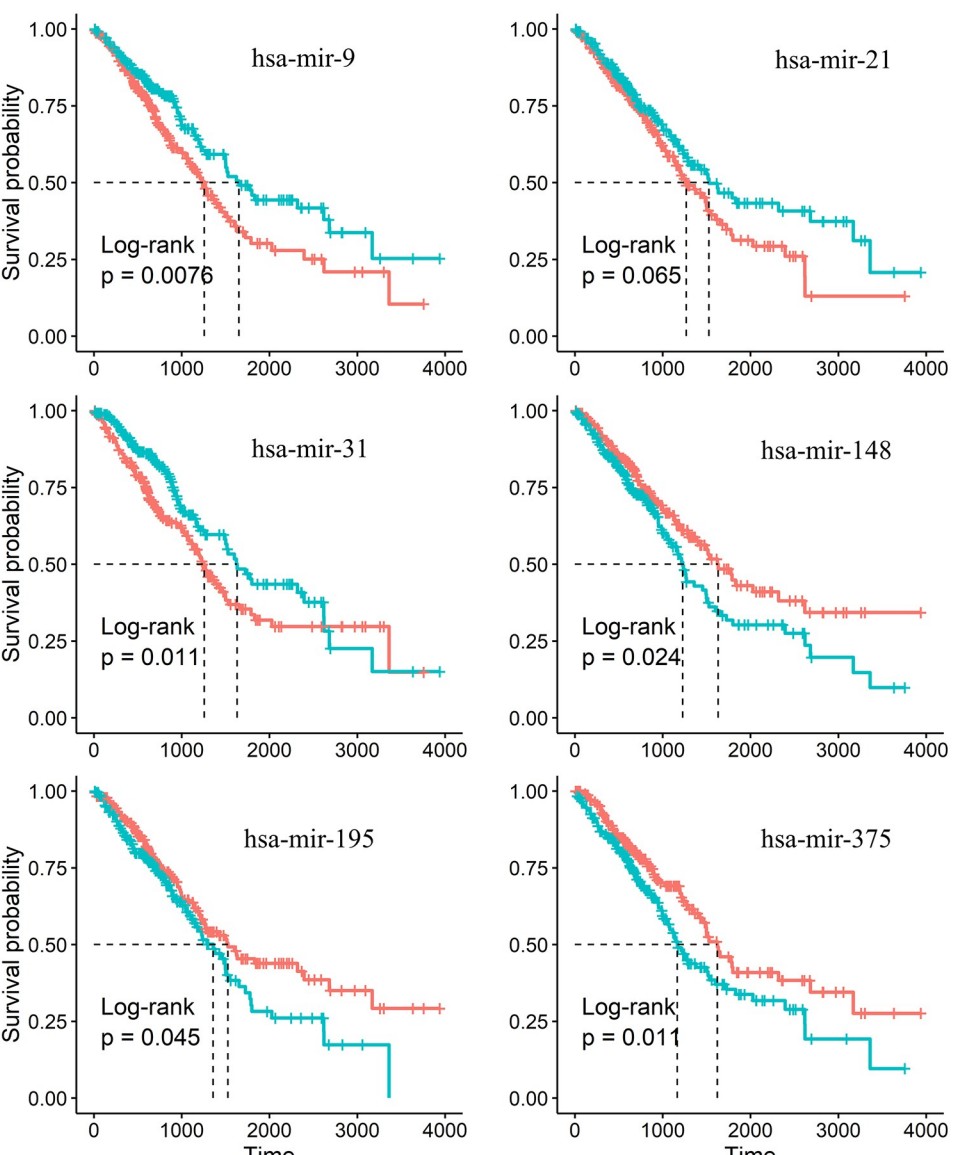

**Fig 7. Survival curves of microRNAs from Kaplan-Meier analysis.** Six microRNAs in CCEN that have a significant effect on the survival of LUAD patient survival in all four stages. Log-rank tests are used to analyze the Kaplan-Meier survival curve.

## Conclusion

LUAD has more pronounced genomic variations [40] than other lung cancer subtypes, which are rarely caused by a few genetic changes, rendering necessary articulating alternative methodologies in order to obtain a reasonable understanding of the complicated mechanisms behind the evolution of LUAD. Analyses based on the CeRNA hypothesis provide a promising approach to understanding mutual regulation at the post-transcriptional level [9, 10, 41].

We have reconstructed the CeRNA networks for lncRNAs, microRNAs and mRNAs corresponding to the four stages of LUAD. Support for the validity of the microRNA-associated CeRNA networks is obtained by comparing variations in the fold-change values and expression levels, as well as by enriching mRNAs to cancer-related items in gene ontology and

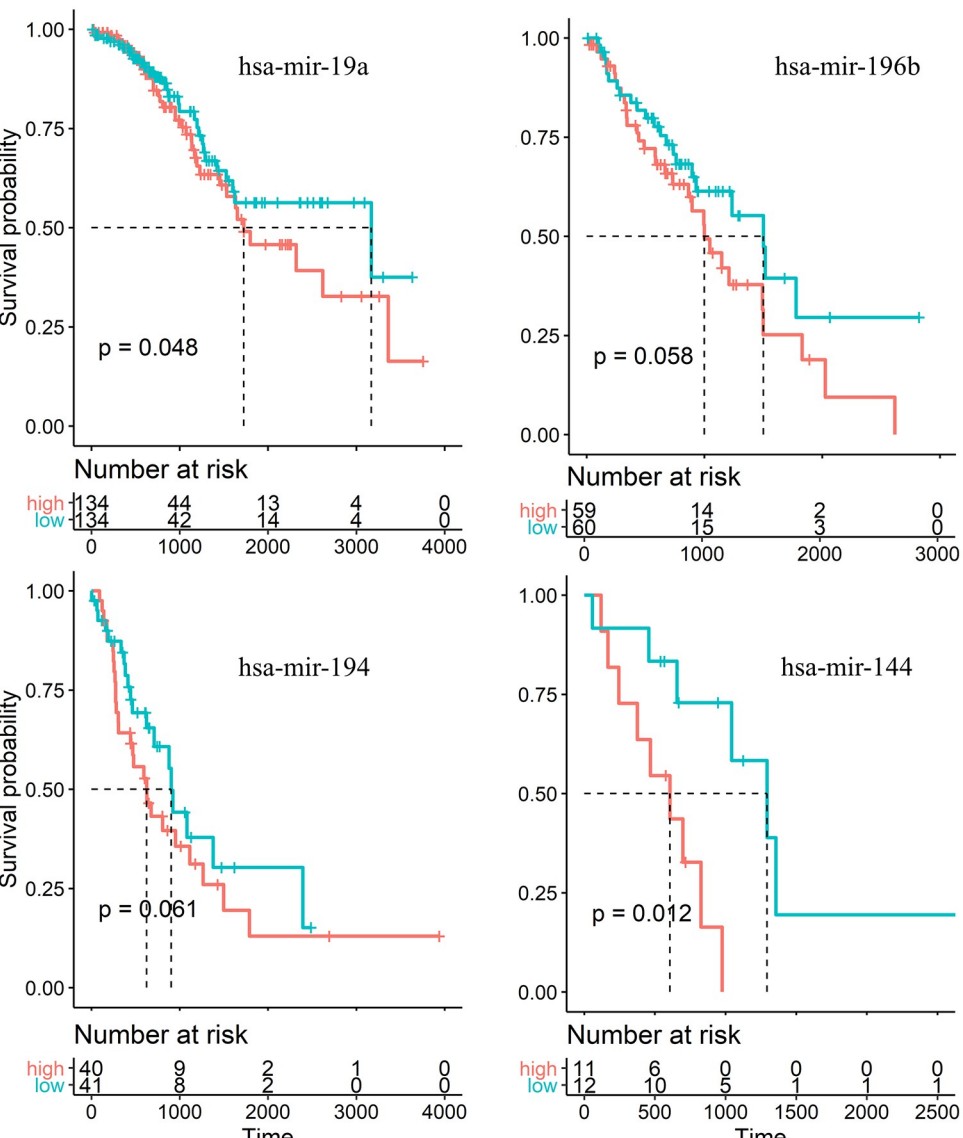

**Fig 8. Survival curves of microRNAs from Kaplan-Meier analysis.** Four microRNAs in the CeRNA networks, each affecting the survival of a specific stage. Log-rank tests are used to analyze the Kaplan-Meier survival curve.

KEGG pathways. Through a systematic enrichment efficiency analysis on mRNAs linked by microRNA in the UCENs and mRNAs selected directly from CeRNA networks (Fig 6), we have identified the key underpinning RNA molecules for LUAD. Especially, by partitioning the network, we find that some specific microRNAs in the CCEN and UCENs have significant influence on the survival rate of LUAD patients (Figs 7 and 8). By performing KEGG pathway enrichment analysis on the mRNA of the CeRNA networks, we find that these mRNAs are not only related to cancer pathways as well as cell cycles and microRNAs in cancer, but there is also a reduction in the degree of the corresponding correlation as the patient state deteriorates from stage I to stage IV of LUAD. This is strong indication that these mRNAs and their CeRNA networks are involved in LUAD and its evolution.

Through the Kaplan-Meier survival analysis [39], we find six microRNAs that markedly affect the patient survival rate for all four stages of LUAD. The first is hsa-mir-9 reported in recent studies, which is involved in the regulation of NSCLC-related eukaryotic translation initiation factor 5A2, TGFBR2, Lamins and other biomolecules [42–44]. Another microRNA is hsa-mir-21, the first oncogenic microRNA discovered [45], which is associated with pulmonary fibrosis [46], inhibits PTEN, promotes growth and invasion [47], affects cell proliferation, migration, and survival [48, 49]. The third one is hsa-mir-31, which is involved in the inhibition of specific tumor suppressants in human lung cancer [50], affects the survival of LUAD patients [51] and participates in the regulation of lung cancer treatment [52, 53]. The fourth one is hsa-mir-148a, which inhibits the metastasis, proliferation and pathogenesis of NSCLC [54–56]. The fifth one is hsa-mir-195, which participates in the process of affecting lung cancer by regulating MYB, cyclin D3, Ailanthone [57–59]. The sixth one is hsa-mir-375, which affects YAP1 molecule [60] in lung cancer and is involved in the regulation of multiple lung cancer stages [61] as a target of Claudin-1 in NSCLC [62]. It is also involved in mouse alveoli inhibition of Wnt/b-catenin pathway [63]. These previous studies suggest that the six microRNAs that we have successfully identified through our quantitative analysis of the CeRNA networks of LUAD are indeed significant for cancer development, validating and demonstrating the power of our framework of dynamical network analysis.

We have also identified four microRNAs that affect the survival of patients at a specific stage of LUAD. In particular, hsa-mir-19a affects the survival at the first stage, which regulates biological molecules such as grape seed procyanidin, MTUS1, c-Met, and FOXP1 to affect lung cancer [64–67]. MicroRNA hsa-mir-196b influences the survival of patients at the second stage of LUAD, which regulates targets Homeobox A9 and Runx2, etc. [68, 69] and acts on early stage lung cancer [70]. MicroRNA hsa-mir-194 has an effect on the survival of patients at the third stage of LUAD, which can serve as a target for human nuclear distribution protein C [71] and constitute a negative feedback loop with Cullin 4B in carcinogenesis [72]. Finally, hsa-mir-144 impacts the patient survival at the fourth stage of LUAD, which regulates EZH2, TIGAR, Lico A, etc. that affect lung cancer [73–75].

The emerging field of network physiology and medicine [23–27] focuses on the associations between network structures and physiologic states, which relies heavily on measurements from biomedical experiments on large spatial and temporal scales. In this sense, our findings demonstrate an association between CeRNA network and physiologic dysfunction of the lung organ.

While CeRNA network analysis can play a pivotal role in the study of post-transcriptional gene regulation in cancer and disease treatment, there are limitations. For example, for the analysis to be valid, the relative concentrations of CeRNAs in the cellular microenvironment should not be too heterogeneous. There can also be defects and errors in the existing microRNA targets data bases.

Another issue concerns the use of possible surrogate test, which is useful for assessing the effects of different data sources on the results. However, our work is to analyze the CeRNA network constructed from genomics data collected from TCGA and clinical data from a large number of LUAD samples in a step-by-step manner. If we use surrogate test to obtain paired signals from different LUAD samples to construct the CeRNA network, there will be many possible CeRNA networks, making it extremely difficult (if not impossible) to determine the networks for identifying biomarkers. Another aspect of our work is a focus on the variations among the progressive stages of lung cancer, and the number of sequential CeRNA networks constructed for different LUAD stages is limited. Our approach is then to use the gene expression, the FC values of the CeRNA network nodes, and gene enrichment analysis to validate the network. For surrogate networks, the meanings of these tests are not clear. We have

demonstrated that our approach does lead to a number of microRNA biomarkers, and we have carried out indirect validation to test the effectiveness of the targets, which includes the Kaplan-Meier survival analysis based on combining the time event correspondence to the clinical physiological data of the sample patient and *a priori* knowledge or expert system verification.

Taken together, based on the CeRNA hypothesis, we have developed a framework to reconstruct the CeRNA networks for the four evolutionary stages of LUAD. Our analysis has yielded some prognostic markers closely related to the survival of LUAD patients. The comparative study of the CeRNA networks for the four stages of LUAD provides new insights into understanding cancer mechanisms and identifying targets for better drugs.

## Materials and methods

### Clinical and gene expression data of LUAD

A total of 877 samples with the corresponding lncRNA, microRNA and mRNA expression profile data from the cancer genome atlas (TCGA) were used. Especially, the numbers of lncRNA, microRNA and mRNA profiles are 29095, 1881 and 25527, respectively. Our analysis requires pre-processing to match the RNA expression data with the patient clinical data, which has excluded 269 samples. We follow the Tumor-Node-Metastasis (TNM) staging criteria for malignant tumors to classify LUAD into four stages. In particular, TNM is a standardized classification system established by the International Association for the Study of Lung Cancer to describe the development of lung cancer in terms of size and spread, where "T" describes the size of the tumor and any spread of the cancer into nearby tissues, "N" denotes the spread of the cancer to nearby lymph nodes, and "M" stands for metastasis, i.e., the spread of the cancer to other parts of the body [76]. For our datasets, we have that, of the remaining 608 samples, 288, 124, 85, and 26 are labeled as stages I, II, III, and IV, respectively, whereas 85 are normal. (More details about the classification criteria of LUAD can be found in S1 Appendix). We perform the match pre-processing on the gene expression profiles as well, resulting in 14460, 1881 and 24991 lncRNA, microRNA and mRNA profiles, respectively. We integrate and further match the clinical physiological data of the samples with either the lncRNA and mRNA data or the microRNA data. For match with the lncRNA and mRNA data, we have a total of 575 cases, where 26, 284, 122, 84, and 59 samples belong to stages I, II, III, IV, and normal, respectively. For match with the microRNA data, there are 24, 279, 122, 85, and 46 samples corresponding to stages I-IV LUAD and normal cases, respectively. (More details are listed in S2 Table).

### Framework of analysis

Our articulated framework of dynamical network analysis combines the following methods of analysis: differential expression analysis, match with microRNA-mRNA and microRNA-lncRNA interaction data, identification of negative correlation between microRNA-mRNA and microRNA-lncRNA expressions, CeRNA network partition, gene ontology and KEGG enrichment analysis, and survival analysis. A flow chart of these methods is illustrated in Fig 9. In the following, each of the methods is described.

### Differential expression analysis

We use the R-language "Limma" package [77] to analyze the differentially expressed lncRNA, microRNA and mRNA profile data [78] in the four stages of LUAD, with the threshold of $\log_2 FC$ (log2 fold change) absolute value for filtering the three types RNAs set as one, while ensuring that their P-value (t-statistic [79], see S2 Appendix for detail) is less than 0.05. Fold changes

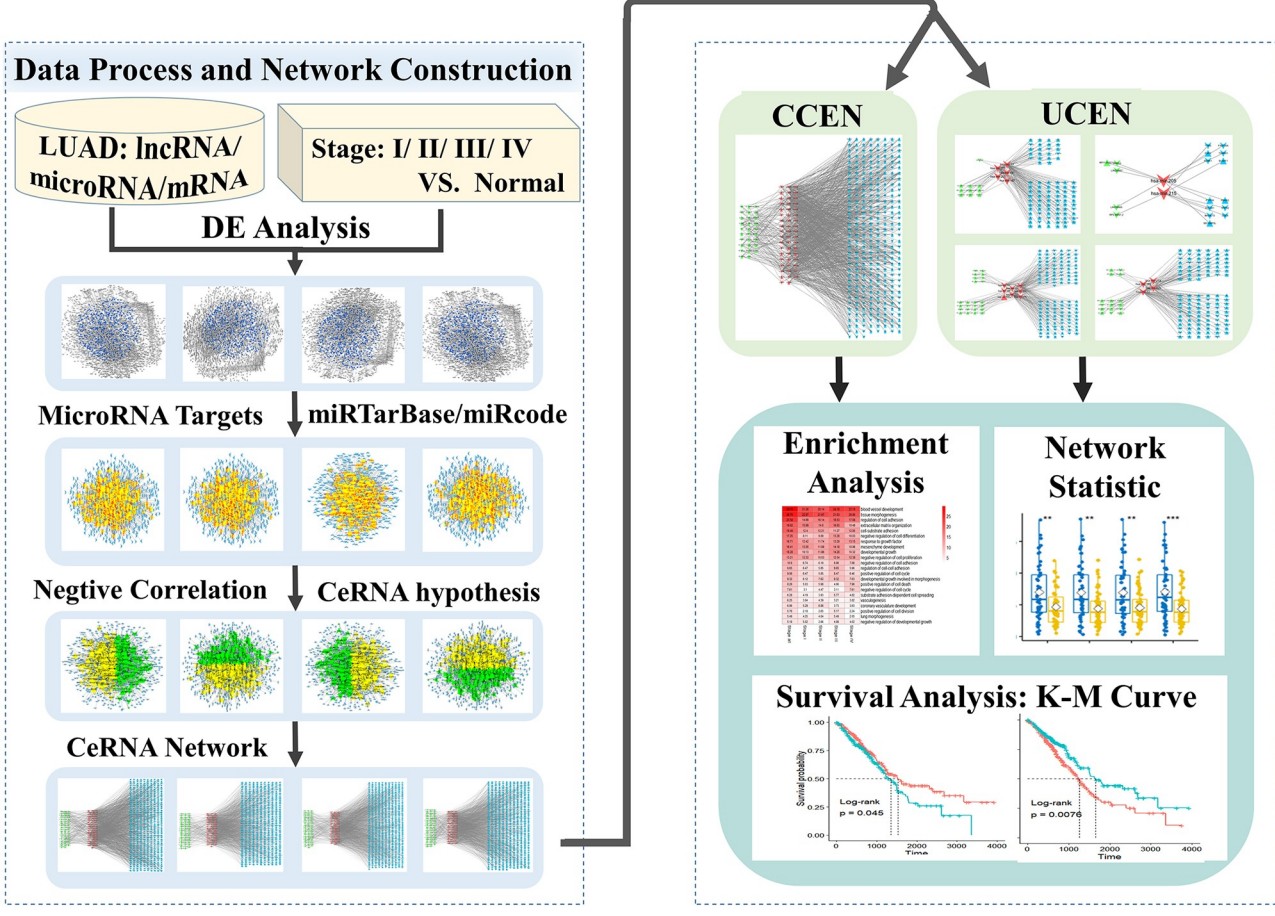

**Fig 9. Overall work flow chart of proposed framework of dynamical network analysis of lung cancer.** LUAD samples are divided into four stages according to the TNM criteria. Differential expression gene analysis and target selection based on negative correlation are performed on the gene expression data of lncRNAs, microRNAs and mRNAs to construct the corresponding CeRNA network for each of the four LUAD stages. The CeRNA networks are divided into UCENs and CCEN. A statistic analysis of the network nodes as well as gene enrichment and survival analyses are carried out.

greater than one correspond to an up-regulated gene, while those less than minus one are associated with a down-regulated gene. We then obtain the numbers of lncRNAs, microRNAs and mRNAs and the up or down regulated expression genes in the four stages of LUAD, as listed in S3 Table.

The individual RNA behavior can be assessed from the volcano map of RNA expression. Fold Change characterizes the relative expression level of interested samples to that of the control samples. For $\log_2$ Fold-Change = 1, the individual RNA expression level equals the group average of RNAs. We have provided the volcano maps for the expressions of lncRNA (S5 Fig), microRNA (S6 Fig) and mRNA (S7 Fig) at the four LUAD stages.

## Data of microRNA-mRNA and microRNA-lncRNA interactions

The base of complementary pairing matching relationship data in the RNA sequences for microRNA-mRNA and microRNA-lncRNA are downloaded from the websites of TarBase (version 6.0) [80], miRTarBase (version 6.1) [81], and miRecords (version 4) [82]. The differentially expressed microRNAs are matched with the microRNA-mRNA and microRNA-lncRNA interaction relationships, following which the lncRNAs and mRNAs targeted by these

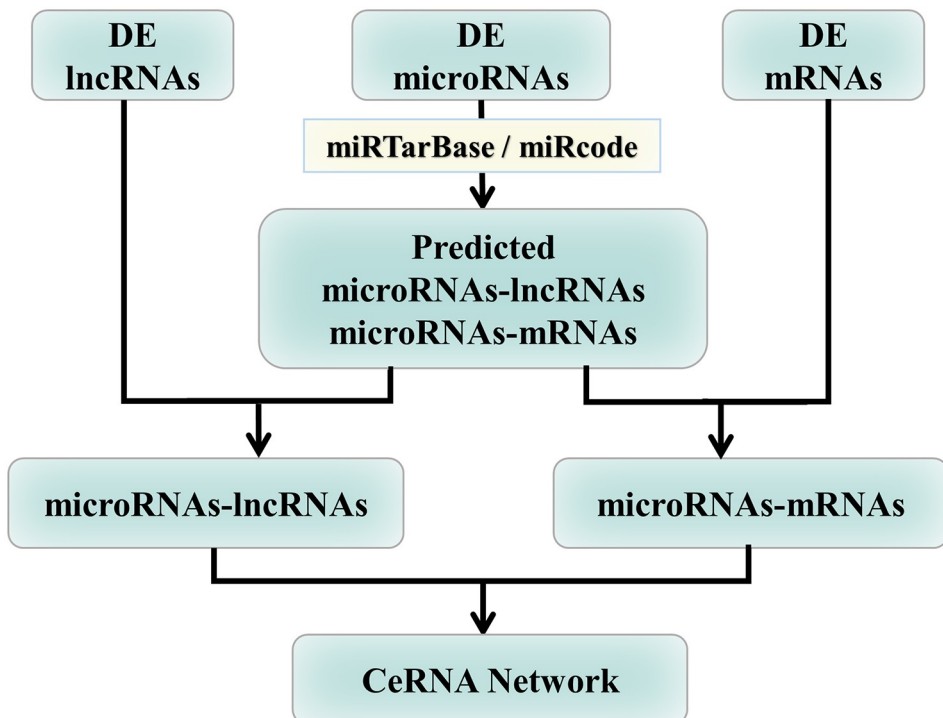

**Fig 10. Selection of targets of microRNAs matched lncRNA and mRNA gene expression data.** Differentially expressed microRNAs are matched to microRNAs of the miRTarBase and miRcode data bases to identify the lncRNAs and mRNAs of the microRNAs targets, which are then matched to differentially expressed lncRNAs and mRNAs, yielding the relationships between microRNAs-lncRNAs and microRNAs-mRNAs and, consequently, leading to the lncRNAs-microRNAs-mRNAs CeRNA networks for the four stages of LUAD.

microRNAs are selected for matching the differentially expressed lncRNAs and mRNAs. A detailed workflow illustrating the analysis of microRNA-mRNA and microRNA-lncRNA interactions is shown in Fig 10, with more details in S4 Table.

## Negative correlation between microRNA-mRNA and microRNA-lncRNA expressions

Based on the CeRNA hypothesis [17, 18], we pick out only those microRNA-mRNA and microRNA-lncRNA interaction pairs with negative correlation calculated from the Pearson correlation of the RNA expression data, where the threshold in the correlation coefficient is set to be -0.1. (See S5 Table for more details).

## Partition of CeRNA network

We select the identical microRNAs among the LUAD four stages as well as the mRNAs and lncRNAs connected by these microRNAs in the CeRNA network as constituting the CCEN, where the numbers of identical lncRNAs, microRNAs and mRNAs emerging in the LUAD CeRNA networks of all four stages are 29, 53 and 283, respectively.

Correspondingly, we compare the one-of-a-kind microRNAs of a specified LUAD stage with those in other stages as well as the mRNAs and lncRNAs connected by those microRNAs in the CeRNA networks, which constitute the UCEN. The numbers of (lncRNAs, microRNAs, mRNAs) in the UCEN networks associated with stages I-IV are (27, 5, 197), (30, 2, 236), (41, 8, 312), and (36, 4, 335), respectively.

## Gene ontology and KEGG enrichment analysis

We perform gene ontology and KEGG enrichment analysis for mRNAs of the LUAD CeRNA networks, taking into account the various biological processes associated with gene ontology.

Gene enrichment analysis is a widely used approach to identifying biological connections. In Figs 3 and 5, we implement the hypergeometric model to assess whether the number of selected genes in the CeRNA networks associated with lung cancer is larger than that which can be expected purely by chance. In particular, the P-value determines whether any term annotates a specified list of genes at frequency greater than that which can be expected by chance, as determined by the hypergeometric distribution:

$$p = 1 - \sum_{i=0}^{k-1} \frac{\binom{N-M}{n-i}\binom{M}{i}}{\binom{N}{n}}, \tag{1}$$

where $N$ is the total number of genes in the background distribution, $M$ is the number of genes within that distribution that are annotated (either directly or indirectly) to the node of interest, $n$ is the size of the list of genes of interest, and $k$ is the number of genes within that list which are annotated to the node. The background distribution by default is all the genes that have an annotation. Then, the outcome of a statistical hypothesis test for the hypergeometric distribution in the gene enrichment analysis, divided by the total number $N$ of genes in the analysis, defines the enrichment efficiency $\eta$:

$$\eta = -\log_{10} P/N. \tag{2}$$

## Survival analysis

We use the Kaplan-Meier method [39], a non-parametric method, to estimate the survival probability from the observed survival data. The survival probability as a function of time is calculated according to

$$S(t_i) = S(t_{i-1})(1 - d_i/n_i), \tag{3}$$

where $n_i$ is the number of patients who were alive before time $t_i$ and $d_i$ is the number of death events at $t_i$. The estimated probability is a step function that changes value only at the time of each event.

Log-rank tests are used to carry out a univariate analysis of the Kaplan-Meier survival curve, which belong to the chi-square test, where all time points of the sample survival information are equal (i.e., with weight setting to one).

## Computational packages

We carry out data processing and statistical analysis using R language (v.3.5.1), analyze differentially expressed RNAs by using the "Limma" package [77], and plot the heatmap of the enrichment analysis and the Kaplan-Meier survival curves using the "heatmap," "survival," and "survminer" packages. The various CeRNA networks are visualized via Cytoscape (v3.6.1). Finally, we perform the gene ontology and KEGG functional enrichment analysis using the online tool Metascape (http://metascape.org).

## Supporting information

**S1 Appendix. TNM stages.**
(PDF)

**S2 Appendix. Statistical analysis and P-value.** We have used four methods of statistical analysis involving hypothesis testing to calculate the P-values.
(PDF)

**S1 Fig. Survival curves of hsa-mir-19a.** The P-values of hsa-mir-19a in the Kaplan-Meier survival curves are 0.24, 0.95, 0.13 for stages II, III and IV, respectively. Log-rank tests are used to analysis of Kaplan-Meier survival curve.
(TIF)

**S2 Fig. Survival curves of hsa-mir-196b.** The P-values of hsa-mir-196b in the Kaplan-Meier survival curves are 0.58, 0.7, 0.99 for stages I, III and IV, respectively.
(TIF)

**S3 Fig. Survival curves of hsa-mir-194.** The P-values of hsa-mir-194 in the Kaplan-Meier survival curves are 0.87, 0.92, 0.6 for stages I, II and IV, respectively.
(TIF)

**S4 Fig. Survival curves of hsa-mir-144.** The P-values of hsa-mir-144 in the Kaplan-Meier survival curves are 0.9, 0.35, 0.58 for stages I, II and III, respectively.
(TIF)

**S5 Fig. Individual lncRNA expressions.** Volcano map of the lncRNA expression level of four stages of LUAD samples. The $x$-axis is $\log_2 Fold-Change$ (FC value), and the $y$-axis is the P-values from the differential expression analysis.
(TIF)

**S6 Fig. Individual microRNA expressions.** Volcano map of the microRNA expression level of four stages of LUAD samples.
(TIF)

**S7 Fig. Individual mRNA expressions.** Volcano map of the mNA expression level of four stages of LUAD samples.
(TIF)

**S1 Table. The number of nodes and edges of CeRNA networks.** Detailed parameter values of the reconstructed lncRNA-microRNA-mRNA CeRNA networks.
(PDF)

**S2 Table. Distribution of clinical samples in gene expression data of LUAD.** Clinical physiological data of the four LUAD stages matched with the lncRNA and mRNA expression data.
(PDF)

**S3 Table. The number of differential expression (DE) gene.** Differentially expressed lncRNA, microRNA and mRNA profile data in the four stages of LUAD.
(PDF)

**S4 Table. The number of lncRNAs or mRNAs targeted by microRNAs and their relationships.** The microRNA-mRNA and microRNA-lncRNA interactions relationships obtained by using the base of complementary pairing matching relationship data in the RNA sequences.
(PDF)

**S5 Table. The number of lncRNAs or mRNAs selected by introducing negative correlation.** The Pearson correlation of the RNA expression data between microRNA-mRNA and microRNA-lncRNA expressions is calculated.
(PDF)

**S1 Data. Figure data.**
(RAR)

## Acknowledgments

We thank Prof. Celso Grebogi and Prof. Lei Yang for helpful discussions.

## Author Contributions

**Conceptualization:** Zhi-Tong Bing, Bing-Hui Guo, Liang Huang, Zi-Gang Huang, Ying-Cheng Lai.

**Data curation:** Chao Kong, Zhi-Tong Bing.

**Formal analysis:** Chao Kong, Zhi-Tong Bing, Bing-Hui Guo, Zi-Gang Huang.

**Funding acquisition:** Zhi-Tong Bing, Bing-Hui Guo, Zi-Gang Huang.

**Investigation:** Chao Kong, Yu-Xiang Yao, Zhi-Tong Bing.

**Methodology:** Yu-Xiang Yao, Zhi-Tong Bing, Zi-Gang Huang.

**Project administration:** Zhi-Tong Bing, Bing-Hui Guo, Zi-Gang Huang.

**Resources:** Zhi-Tong Bing, Bing-Hui Guo, Liang Huang, Zi-Gang Huang.

**Validation:** Chao Kong, Zi-Gang Huang, Ying-Cheng Lai.

**Visualization:** Chao Kong.

**Writing – original draft:** Chao Kong.

**Writing – review & editing:** Yu-Xiang Yao, Liang Huang, Zi-Gang Huang, Ying-Cheng Lai.

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
