## [Decision Letter · Decision Letter 0]

5 Feb 2020

Dear Dr Huang,

Thank you very much for submitting your manuscript "Dynamical network analysis reveals key microRNAs in progressive stages of lung cancer" for consideration at PLOS Computational Biology. As with all papers reviewed by the journal, your manuscript was reviewed by members of the editorial board and by several independent reviewers. The reviewers appreciated the attention to an important topic. Based on the reviews, we are likely to accept this manuscript for publication, providing that you modify the manuscript according to the review recommendations.

Sincerely,

Ilya Ioshikhes

Associate Editor

PLOS Computational Biology

Douglas Lauffenburger

Deputy Editor

PLOS Computational Biology

[LINK]

Reviewer's Responses to Questions

**Comments to the Authors:**

Reviewer #1: An important discovery in cancer research is the roles played by non-coding RNAs (ncRNAs) that have been identified as the oncogenic drivers and tumor suppressors in different cancer types. This has led to the fundamental competing endogenous RNA (CeRNA) hypothesis: microRNAs, mRNAs, and ncRNAs form an inseparable unity of RNA-level regulating network in the intracellular environment associated with cancer development, collectively known as the CeRNA network. In existing studies of CeRNA networks, a common approach was to identify some differentially expressed lncRNAs, mRNAs and microRNAs based on absolute fold changes and the values of the false positive ratio. A deficiency of this approach is that information used to construct the underlying CeRNA networks and to reveal the network functions with clinical implications is directly from data bases without consideration of any dynamical aspect. As a result, the reconstructed CeRNA networks are simply a combination of different kinds of RNAs, whereas the dynamical interplay and competition among different types of RNAs are completely ignored. However, cancer development and evolution are fundamentally a dynamical process. Reflected in the underlying CeRNA networks, it is unlikely that the intrinsic interactions, interplay and the network structure remain static during cancer development. To better understand cancer and to identify more effective biomarkers, the dynamical aspects of the CeRNA networks need to be studied. In the manuscript under review, the authors developed a dynamical network analysis to address this issue. Through this analysis, the authors uncovered ten specific microRNAs that can serve as effective biomarkers and prognostic tools for lung cancer.

With available data from a gene database, e.g., RNA expression and clinical data of LUAD from the cancer genome atlas, the authors incorporated the time axis into the analysis by focusing on the four stages of LUAD progression to construct the lncRNA-microRNA-mRNA CeRNA network for each stage. The quantitative approach to reconstructing the CeRNA network consists of analyzing differentially expressed RNAs, matching microRNA targets by base complementary pairing, and selecting the negative correlation by invoking the CeRNA hypothesis. The authors then calculated the fold changes and the average expression level of RNAs in the CeRNA networks for the four stages and carried out Gene Ontology and Kyoto Encyclopedia of Genes and Genomes pathway enrichment analyses to validate the results. A finding is the emergence of two characteristically distinct types of networks that play an important role in the progression of LUAD from stage I to stage IV: one is common to all four stages, which the authors named as the common competing endogenous RNA network (CCEN), and another is unique for each stage, which the authors termed as the unique competing endogenous RNA networks (UCENs). Analyzing the properties of CCEN and UCENs, the authors uncovered a number of key genes that affect or even determine the survival of patients at each stage of LUAD: six microRNAs from the CCEN which affect the survival of LUAD patients at all stages, and four other microRNAs that influence the survival at each specific stage.

The work represents a timey contribution to cancer research because it has established a more comprehensive gene-data analysis framework than previous ones, not only providing a tool to probe more deeply into the mechanisms of cancer evolution than previously possible but also having the potential to lead to more effective biomarkers and drug targets for LUAD as well as other types of cancer. The paper is very well written. I recommend publication in its present form.

Reviewer #2: The authors construct and analyze CeRNA networks for four sequential stages of lung adenocarcinoma (LUAD) based on multi-omics data of long non-coding RNAs (lncRNAs), microRNAs and mRNAs. Their analysis shows the emergence of two characteristically distinct types of networks, common competing endogenous RNA network (CCEN) and unique competing endogenous RNA networks (UCENs), that play an important role in the progression of 4 different stage LUAD. The study yields some prognostic markers closely related to the survival of LUAD patients, and provides new insights into understanding cancer mechanisms and identifying targets for better drugs. The results are novel and interesting.

There are several points which authors should address and comment on before publication:

1. The authors study the CeRNA networks in four LUAD stages. The author should introduce some background on how these four stages are classified and what are the major differences among these four stages.

2. In figure 2, blue dots represent Gene expression values and yellow dots represent differential expression analysis results. However, it is not clear what does the legend “Retention“ and “Discard” means in the figure. It is not described neither in the figure caption nor in the main text. The author should give explanations to it.

3. Since the study have utilized statistical tests and obtained P-values for different tests, the authors should describe what type of tests are used during different analyses in the Methods section.

4. The authors calculate the average expression levels of RNAs and CeRNA networks. The authors should also provide a few individual behaviors in the Supplementary Material to show how different is the individual behavior from the group average.

5. It would be instructive to do some additional surrogate test to prove the results are physiologically meaningful. For example, the authors could make a CeRNA network consists of RNAs from different subjects, or to compare networks from real data with those from noisy/artificial processes, and check whether reported behavior is statistically different from the surrogate tests.

6. The authors studied the CeRNA networks in four LUAD stages and find the emergence of two characteristically distinct types of networks play important roles in the progression of 4 different stage LUAD. The findings of transitions in CeRNA networks with progression of the stage/disease is reminiscent of transitions and reorganization in network of physiological /organ systems interaction in the human body at larger space/time scales, reported in earlier studies within the broader context of the field of Network Physiology – see for example:

-Bashan A, et. al. Network physiology reveals relations between network topology and physiologic function. Nature Communications 2012; 3: 702 doi: 10.1038/ ncomms1705.

-Bartsch RP, et. al. Network Physiology: how organ systems dynamically interact. PLOS ONE, 2015, 10(11): e0142143

-Liu KKL, et. al. Plasticity of brain wave network interactions and evolution across physiologic states. Frontiers in Neural Circuits, 2015; 9:62

-Liu KKL, et. al. Major component analysis of dynamic networks of physiologic organ interactions. Journal of Physics: Conference Series, 2015; 640, 012013

-Ivanov PCh, et. al. Focus on the emerging new fields of network physiology and network medicine. New Journal of Physics, 2016; 18:100201.

It would be instructive if the authors make parallels and discuss their results within the framework/context of Network Physiology, and the above works.

**Have all data underlying the figures and results presented in the manuscript been provided?**

Reviewer #1: Yes

Reviewer #2: None

PLOS authors have the option to publish the peer review history of their article (what does this mean?). If published, this will include your full peer review and any attached files.

Reviewer #1: No

Reviewer #2: No
---

## [Decision Letter · Decision Letter 1]

17 Mar 2020

Dear Dr Huang,

We are pleased to inform you that your manuscript 'Dynamical network analysis reveals key microRNAs in progressive stages of lung cancer' has been provisionally accepted for publication in PLOS Computational Biology.

Best regards,

Ilya Ioshikhes

Associate Editor

PLOS Computational Biology

Douglas Lauffenburger

Deputy Editor

PLOS Computational Biology

Reviewer's Responses to Questions

**Comments to the Authors:**

Reviewer #2: The authors have addressed all my points, and I suggest publication.

**Have all data underlying the figures and results presented in the manuscript been provided?**

Reviewer #2: None

PLOS authors have the option to publish the peer review history of their article (what does this mean?). If published, this will include your full peer review and any attached files.

Reviewer #2: No

---

## [Editor Report · Acceptance letter]

1 May 2020

PCOMPBIOL-D-19-02215R1 

Dynamical network analysis reveals key microRNAs in progressive stages of lung cancer

Dear Dr Huang,

I am pleased to inform you that your manuscript has been formally accepted for publication in PLOS Computational Biology. Your manuscript is now with our production department and you will be notified of the publication date in due course.

With kind regards,

Laura Mallard
